# Addressing the Scarcity of Traditional Water Sources through Investments in Alternative Water Supplies: Case Study from Florida

**Tatiana Borisova [1,\*], Matthew Cutillo [2], Kate Beggs [2] and Krystle Hoenstine [2]**

[1]   Food and Resource Economics Department, Institute of Food and Agricultural Sciences, University of Florida, Gainesville, FL 32611, USA

[2]   Economic and Demographic Research, Florida Legislature, Tallahassee, FL 32399, USA; cutillo.matthew@leg.state.fl.us (M.C.); beggs.kate@leg.state.fl.us (K.B.); krystle.hoenstine@gmail.com (K.H.)

**\***   Correspondence: tborisova@ufl.edu

**Abstract:** This paper examines the capital costs for alternative water supply projects in Florida, the third most populous state in the United States. The increasing scarcity of fresh groundwater in Florida has led to investments in alternative water supply sources, including brackish groundwater, surface water capture and storage, reclaimed water, and stormwater. Expenditures to meet the growing water demand for the 20-year planning horizon are estimated using water demand projections and existing supply estimates from Florida's five water management districts. In the regions where demand projections exceed the existing supply, the districts are required to identify project options to meet the growing water demand while protecting the natural systems. This study uses the database of 645 projects implemented in the past or considered for the future. The Ordinary Least Squares regression model shows that project implementation costs depend on project capacity, type, implementation status, and implementation region. Given the most common project types and project sizes, the total investments to meet the state's future water demand could reach almost $2 billion in the next 20 years. The expenditures necessitate more cost-effective options (such as expanding stormwater use and water conservation).

**Keywords:** long-term; regional water supply planning; alternative water supply; projects; expenditures; investments; conservation

---

## 1. Introduction

Expenditure forecasting is an integral part of long-term regional water supply development and water resource management [1]. In many regions, a growing population, expanding economy, and propagating groundwater depletion have increased water scarcity and impacted water-dependent ecosystems [2–5]. To address communities' water supply and environmental protection priorities, water suppliers and government agencies seek to diversify water supply sources, invest in demand management, and implement environmental protection strategies [1,6]. Due to the high costs of such initiatives, they require funding from a mix of local, regional, state, and federal sources [7,8]. Government cost-share and private–public partnerships have become common funding strategies for water infrastructure projects, supplementing funding from service fees, bonds, and other funding sources available to water utilities. Infrastructure funding from the general government revenue has been justified as a method to promote economic development, given that infrastructure is a prerequisite of economic growth [7,8]. Despite the seemingly diverse funding opportunities, funding sources are becoming less available. Economic recessions and utility revenue shrinkages due to per capita water use reduction translate into a funding deficit for water supply development and environmental

protection. Therefore, forecasting expenditures and funding needs have grown in importance in long-term regional water supply planning on the local, regional, and state levels.

This paper aims to estimate the expenditures required to develop alternative water supplies and meet the water demand increase in Florida, the third most populous state in the United States. Florida's population is expected to grow from almost 22 million in 2020 to over 26 million in 2040 [9]. The increase in population is projected to escalate water demand unless water use efficiency significantly improves [10]. The state's traditional water source—fresh groundwater—is becoming scarce, with withdrawals causing or projected to cause harm to the water resources or ecology in several areas [11]. "Rethinking water supply" [12] (p. 21302) to include the sources that have previously been ignored, along with water conservation, is seen as a solution to water challenges. Assessing the costs and evaluating funding sources for such supply projects become increasingly important in Florida, mirroring the trend observed in other states and countries.

## 2. Study Area, Materials, and Methods

### 2.1. Study Area: Florida's Long-Term Water Supply Planning Framework

Florida has been primarily relying on groundwater, which is referred to as "traditional" water supply source in most of the state. One of the leading groundwater users in the United States, Florida reports more than four billion gallons (that is more than 15 million m$^3$) of groundwater withdrawals daily [13]. In 2015, groundwater accounted for 63.0 percent of total freshwater withdrawals, and it was particularly prominent in the public supply (86.2% of the total freshwater withdrawals in this category) [13]. The groundwater withdrawals exceed the recharge rates in many Florida areas, with reductions in the flows and levels documented for selected aquifers, springs, streams, and lakes. Minimum flows or minimum water levels (MFLs) are defined as the limits at which further water withdrawals would be significantly harmful to the area's water resources or ecology (§ 373.042, Fla. Stat.). As of 1 March 2019, out of 426 adopted MFLs, 105 are in recovery status, implying that their water flows or water levels are below their adopted MFLs [14], and groundwater withdrawals in the relevant basins should be carefully managed or reduced.

To address and prevent conflicts among water users (e.g., see [15,16]), Florida has developed a rigorous framework for long-term water supply planning. The state is divided into five water management districts (WMDs) based on surface water hydrologic boundaries (Figure 1): Northwest Florida (NWFWMD), Suwannee River (SRWMD), St. Johns River (SJRWMD), Southwest Florida (SWFWMD), and South Florida (SFWMD). Each WMD's governing board is required to develop a district water management plan (§ 373.036, Fla. Stat.) for at least a 20-year planning horizon. A principal element of a district's water management plan is a water supply assessment (WSA). The WSA determines whether existing and reasonably anticipated sources of water and conservation efforts are adequate to sustain the water resources and related natural systems into the future. In a region where a WMD has determined that existing water sources are inadequate to meet projected demand, a more in-depth regional water supply plan (RWSP) must be developed for that region. Both districtwide WSAs and RWSPs are required to be updated at least once every five years (§ 373.036, Fla. Stat.).

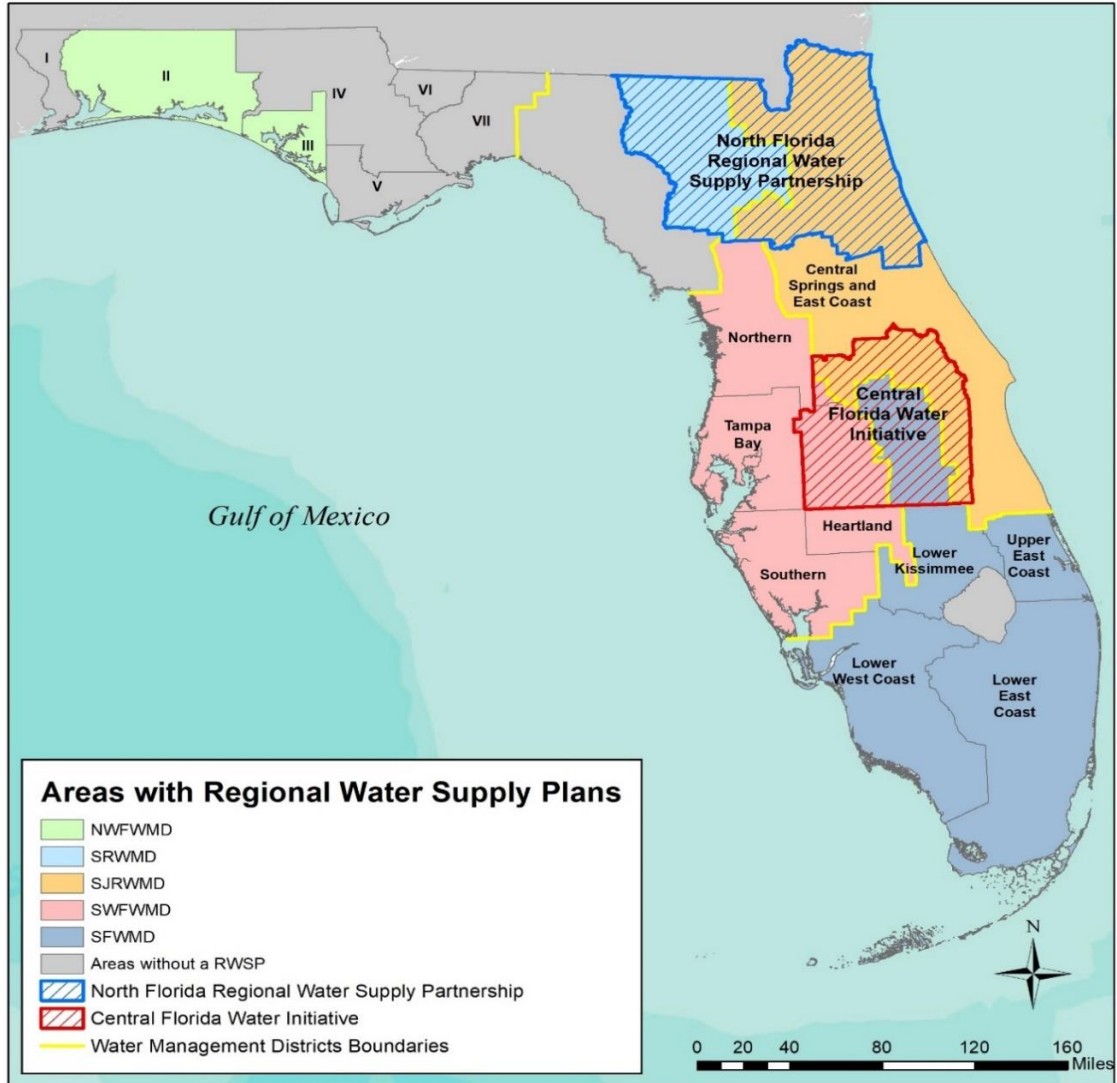

**Figure 1.** Florida's water management districts and water supply planning regions (Source: Originally developed by FDEP, Office of Water Policy, for [17]).

As Figure 1 indicates, RWSPs have been developed for many areas, implying that the projected demand exceeds the existing supplies in most of the state. In two regions, WMDs have cooperatively developed RWSPs that span across WMD boundaries. These are the North Florida Regional Water Supply Partnership (NFRWSP) and the Central Florida Water Initiative (CFWI) established for the regions surrounding Jacksonville (in northeast Florida) and Orlando (in central Florida). In these two areas, populations are booming, and water withdrawals from the shared aquifer are particularly high.

Each RWSP contains a list of water supply development project options and water resource development programs. According to the Florida Statutes, water resource development project options must include projects that support water supply development for all uses and the natural systems in the region (§ 373.709, Fla. Stat.). Examples of such project options are regional model development and data collection. In turn, the water supply development project options include traditional and alternative water supply projects (§ 373.709, Fla. Stat.). The water that can be made available from the projects in an RWSP must exceed the water supply needs in the water supply planning region for all existing and future reasonable-beneficial uses within the 20-year planning horizon (§ 373.709, Fla. Stat.).

The water supply development project options should be technically and financially feasible. Note that unless otherwise exempt, all water withdrawals in Florida are regulated through a system of consumptive use permits granted for a fixed time (e.g., 20 years) by WMDs. The water supply development projects included in RWSPs are referred to as "options:" local governments, public and private utilities, self-suppliers, and other water users may choose from these project options when applying for the new water use permits or an extension or modification of the existing permits. Alternatively, water users can propose different projects. The WMD shall determine whether such proposed projects meet the goals of the RWSP. If so, the projects shall be included in the list of projects supporting RWSP.

An RWSP should also account for water conservation and other demand management measures, as well as water resources constraints. In addition, RWSPs must include projects identified in an applicable recovery or prevention strategy (RPS) for established minimum flow (for rivers, streams, estuaries, and springs) or minimum water levels (for lakes, wetlands, and aquifers) (§ 373.0421, Fla. Stat.).

*2.2. Florida's Long-Term Water Demand and Supply Projections*

The most substantial increase in the statewide demand is expected for the public supply category, representing residential, commercial, and industrial customers served by public and private utilities. The demand is also forecasted to grow in the other water use categories that are not supplied by the water utilities and therefore classified as "self-supplied." Based on [10], these categories are referred to as domestic (e.g., wells providing for both indoor and outdoor household water needs), agriculture, landscape/recreational (e.g., golf courses and parks), commercial/industrial/institutional, and power generation categories (see Figure 2). The demand is generally projected from the five-year average per capita water use for counties or utility service areas and relevant population forecast [17]. For agriculture, an econometric water demand forecasting model utilizes past water withdrawal data, land use trends, and projected agricultural commodity prices [18]. These demand projections follow guidelines for the regional water supply planning [19]. They do not account for future improvements in water use efficiency, potential adjustments in urban development, and other changes that can reduce the future per capita use and per acre irrigation rates. Therefore, these demand projections represent conservative, "status quo" water use forecast.

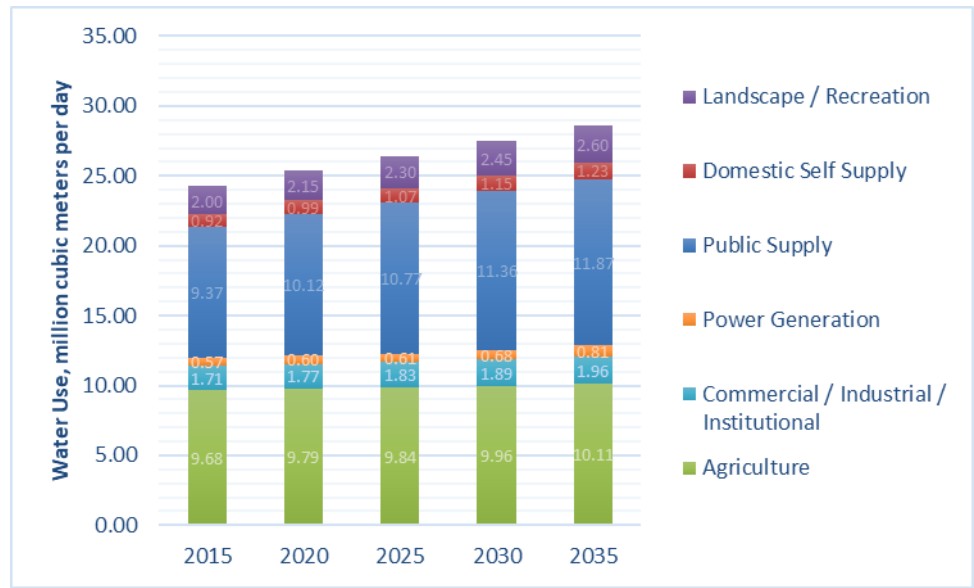

**Figure 2.** Total Statewide Water Use Projections for 2015–2035 Developed by the water management districts (WMDs) for Planning Purposes, assuming average rainfall (Source: based on data from [10]).

Overall, the statewide water demand increase cannot be met solely with the existing sources. WMDs evaluate existing water supplies by using hydrogeological models, examining water utilities' planned projects, comparing existing and permitted water withdrawals, and other methods [17]. The differences between the projected water demand and existing supply are expected to be met through alternative water supplies and conservation.

"Alternative water supplies" are statutorily defined as salt water, brackish surface and ground water, surface water captured predominately during wet-weather flows, surface and groundwater storage, reclaimed water, stormwater, and any other water supply source that is designated as nontraditional in the applicable regional water supply plan (§ 373.019, Fla. Stat.). Reclaimed water refers to "the water that has received at least secondary treatment and basic disinfection and is reused after flowing out of a domestic wastewater treatment facility" (§ 373.019, Fla. Stat.). Alternative water supply sources will play a key role in meeting the increase in water demands. A significant number of options for water supply development are identified in the RWSPs.

Water conservation will also play an essential role in meeting or offsetting the increase in water demand. For planning purposes, water conservation is defined as "the prevention and reduction of wasteful, or unreasonable uses of water to improve the efficiency of use" [19] (p. 30). Districts' conservation projections are intended to represent "reasonably expected demand reduction at the end of the planning period due to conservation activities" [19] (p. 30). The projections are based on agricultural irrigation efficiency trends, residential end-use surveys, statistical evaluation of actual billing data, information from the Water Conservation Tracking Tool [20], and other available data and methods [17].

WMDs consider both passive and active conservation. In the public supply category, passive conservation occurs when fixtures and appliances reach the end of their service life, and they are replaced by new (usually, more efficient) designs. Passive conservation also occurs when newly built homes are more efficient in their water use than the existing homes [21]. Passive conservation can result in substantial per capita water use reduction while having zero costs to the water providers and government agencies. In turn, active conservation accounts for the additional water use reduction due to conservation programs administered by public suppliers, government agencies, Extension, non-governmental groups, and other entities. These programs can include cost-share funds for installing more efficient indoor fixtures, higher water prices, residential irrigation restrictions, outreach strategies, and certification for low water users. Potential reductions in water use due to water conservation are estimated by the WMDs separately from the water demand projections shown in Figure 1. Given that the conservation projections can depend on funding for conservation initiatives, WMDs continue to focus on the "status quo" water demand forecast shown in Figure 2.

The Florida Department of Environmental Protection (FDEP) is charged with providing an annual status summary of regional water supply planning (§ 373.709, Fla. Stat.). The following definitions are used in the summary. "Net Demand Change" is the change in total water demands projected over the planning horizon [19]. "Net Demand Change of which Additional Projects and/or Conservation Must Surpass" is defined as the total water volume needed to meet future demands. It is calculated as the difference between "Net Demand Change" and "Estimated Existing Sources Available to Meet Future Demands" [19]. In turn, "Conservation Projection to Meet Future Demands" refers to the projected conservation savings that could be achieved during the planning horizon [19]. This conservation projection can exceed the values reported in "Net Demand Change of which Additional Projects and/or Conservation Must Surpass." The 2018 status summary, published in August 2019 [10], is compiled in Table 1.

**Table 1.** Projections of the water demand change, existing water sources, and additional water needed for Florida regions (based [10]).

| Water Management Districts and Planning Regions | Net Demand Change (Million Cubic Meters Per Day) | Net Demand Change of which Additional Alternative Water Supply or Conservation Must Surpass (Million Cubic Meters Per Day) | Conservation Projection to Meet Future Demands (Million Cubic Meters Per Day) [1] |
|---|---|---|---|
| Areas planning for 2015–2035 | | | |
| NWFWMD | 0.15 | 0.01 | 0.07 |
| SRWMD & SJRWMD [2] | 0.81 | 0.53 | 0.42 |
| CFWI | 0.88 | 0.88 | 0.14 |
| SWFWMD [2] | 0.66 | 0.13 | 0.37 |
| Total for regions planning to 2035 | 2.50 | 1.55 | 1.01 |
| Areas planning for 2020–2040 | | | |
| SFWMD [2] | 1.76 | 0.24 | 0.54 |
| Statewide | | | |
| Total Statewide | 4.26 | 1.78 | 1.55 [1] |

[1] This column reports the higher of the values provided in [10]. For the lower estimate, the total statewide conservation potential is 1.45 million $m^3$ per day over the 20-year planning period. [2] Excludes areas in CFWI.

As discussed above, the statewide water demand increase cannot be met solely with the existing sources. In the areas with a planning horizon to 2035, the net demand is projected to exceed the existing water supplies by 1.55 million $m^3$ per day (or 408.90 million gallons per day, mgd) by 2035. In the regions planning to 2040, the water demand is expected to surpass the existing water supplies by 0.24 million $m^3$ per day (or 62.58 mgd) by 2040 (Table 1). At the same time, the total water conservation potential statewide is 1.55 million $m^3$ per day (or 408.3 mgd) over the 20 years. To assess the offset for the alternative water supply costs that the water conservation can provide, we compared the water conservation potential, demand increase, and existing supplies for each of the water supply planning regions identified in Figure 1. Conservation potential can offset the requirement to develop water supplies for all planning regions except the northern region in SWFWMD, as well as CFWI, and NFRWSP. The remaining additional water supply needs for these three planning regions are 0.99 million $m^3$ per day (or 260.80 mgd). In other words, water conservation can reduce the necessity for alternative water supplies by almost 45 percent (subject to funding availability for the active water conservation initiatives).

*2.3. Database of Water Supply and Water Resource Development Projects*

This study utilizes a dataset assembled by the FDEP from past project funding information and data from the RWSPs [10]. FDEP annually publishes a project list summarizing the options from current RWSPs along with the projects funded in the past [10]. For implementation costs of completed projects and projects in design or construction, the FDEP dataset includes the "Project Total" column with information about the total project funding by the state (if any), district, or cooperating entity (e.g., county, city, water utility, farm, homeowner association, or golf club). This information is not always reflective of the total implementation cost of the project since it generally does not include information about land purchases or the costs of project components ineligible for cooperative funding from regional or state sources. This information also excludes any funding provided by federal agencies. It is assumed, however, that the funding from the state, district, or cooperating entity accounts for most of the implementation costs. For the projects that are listed as RWSP or RPS options, the "Projected Total Funding (for RWSP/RPS Options Only)" column summarizes information about potential funding requirements (i.e., planning-level cost estimates). This "Projected Total Funding"

is an estimate only and is not verified until the project is submitted for cost-share funding to begin the design or implementation. Still, this projected funding represents the best available information regarding the future funding needs and, therefore, we include it in the analysis. Below, the combined "Project Total" and "Projected Total Funding (for RWSP/RPS Options Only)" is referred to as the "project total ($)". These values are indexed to 2019 USD using the consumer price index [22]. It is important to note that these implementation costs primarily include capital investment needs, while not accounting for operation and maintenance costs [17].

The FDEP list also includes information about project capacity (expressed as water or reuse flow, in mgd). Further, the project status is identified as completed, in design, under construction, on hold, canceled, or "RWSP/RPS Options Only." The projects are classified into 15 types, for example, brackish groundwater, agricultural conservation, reclaimed water (for potable offset), and data collection and evaluation. In addition to the project type, a narrative description is available for each project. Finally, for some of the completed projects, the funding shares of the state, regional, and local funding sources are described.

## 2.4. Regional Project Implementation Scenarios

To estimate statewide expenditures to meet a projected increase in demand, the criteria for identifying the project types and capacities for different planning regions should be defined. Existing studies offer a range of optimization models to assist in the project selection; however, optimization may not serve water planning agencies' needs. Optimization models can consider either a single objective—such as minimizing the total discounted infrastructure investment—or multiple objectives—such as reducing the expenditures while maximizing resilience (e.g., see [6,23–26]). Nevertheless, the number of objectives built into optimization is usually limited, and it may not reflect the full diversity of the stakeholder goals. Water supply planning typically aims at integrating the perspectives of various stakeholder groups, including government agencies, water suppliers, agricultural interests, and environmental nongovernmental organizations, that bring a myriad of considerations to the planning discussions [26,27]. Further, the recommendations based on an optimization model may be perceived by the stakeholders as prescriptive, which may impede reaching an agreement among the diverse interests involved in the planning. For example, in Florida, WMDs rely on water suppliers to identify a range of feasible water supply options. The districts do not have the authority or local knowledge to prescribe specific projects to the suppliers. Therefore, an optimization model may not be suitable for the districts to make water supply recommendations and develop funding estimates on the regional scale. Optimization-based models may also provide inadequate guidance to adaptive planning approaches when the water supply development is expected to evolve as uncertainties are resolved, and additional information becomes available [28].

In this paper, the scenario is based on the description of the potential future water supply sources and the number of projects of each type in the regional water supply plans, as well as informal discussions with the WMDs described in [17]. Specifically, in the FDEP project database [10], the number and the capacity of projects of different types can be calculated, focusing on the projects that have not yet been completed (and therefore, can be used to meet the future water demand). The project number and total capacity vary among the Florida regions. In the expenditure analysis we assume that the share of the different water supply sources should correspond to the share of the projects of different types in the total project capacity for each region. The mean project size is assumed for every project type. The project costs are estimated using a regression model, as discussed below.

## 2.5. Regression Analysis of Project Implementation Costs

The project cost analysis reported in [29] implies a linear relationship between the natural logarithm of the project implementation costs and the natural logarithm of the project capacity. For this study, we assumed a similar linear relationship between the natural logarithms of the costs and capacities. Because factors other than the project capacity can influence implementation costs, a multivariate

regression analysis is used to explore the potential relationship between the project costs and project characteristics:

$$\ln(Cost) = \alpha + \beta \times Type + \gamma \times Region + (\delta \times Status + \zeta \times Type + \eta \times Region) \times \ln(Capacity) + \theta \times \ln(Capacity) + \varepsilon$$

where $\beta$, $\gamma$, $\delta$, $\zeta$ and $\eta$ are vectors of model coefficients; *Status*, *Type*, *Region*, and *Capacity* are vectors of respective project characteristics (i.e., independent variables); and $\varepsilon$ is the error term representing factors other than those captured in the independent variables that influence the dependent variable [28]. In the model, $\alpha$ refers to the intercept, and $\theta$ refers to the model coefficient capturing the relation between the natural logarithms of the project size and project costs. Project size's effect can depend on the project type, location, and status, as captured by the coefficients $\delta$, $\zeta$ and $\eta$. The independent variables *Status*, *Type*, and *Region* are dummy variables that describe the projects' qualitative characteristics; therefore, the intercept $\alpha$ includes information for the benchmark category of *Status*, *Type*, and *Region* used for the comparison with the other categories [30].

The model is first estimated using the ordinary least squares method [30] implemented in the *glmselect* procedure in the SAS 9.4 software (© 2002–2012, SAS Institute, Inc., Cary, NC, USA). Backward selection method with select and stop criteria being the significance level of 0.1 is used to test the selected model specification against the specifications with some of the variables omitted. Furthermore, the model coefficients were compared between the *glmselect* and *robustreg* procedures to account for the potential effect of outliers and high-leverage observations on the model estimation coefficients. Below, the results are reported for the *robustreg* procedure only.

## 3. Results

### 3.1. Water Supply Projects

The original FDEP project dataset included 1623 projects or project phases. After linking all the phases for multi-phased projects, 1417 unique projects were identified. We further removed 302 projects that were canceled or lacked information about the implementation costs or project capacity (leaving 1115 projects in the dataset). Next, 426 projects were excluded since these projects did not directly contribute to the development of new water supply. These projects were described as "data collection and evaluation" (1 project), "flood control works" (4 projects), "agricultural conservation" (63 projects), and "public supply and commercial-industrial-institutional conservation" (358 projects). Discussions with WMD staff allowed us to also exclude desalination (5 projects) and reclaimed water for groundwater recharge or natural system restoration (36 projects) from the expected project types for meeting the future water demands. Finally, three projects were excluded because their type information was not clearly identifiable. The remaining database included 645 projects of seven alternative water supply types (see Table 2).

**Table 2.** Project number, size, and total capacity, by project types.

| Project Type | Number of Projects | Project Size (m³ per day) | | | | Total Water or Reuse Flow for All Projects (m³ per day) |
| --- | --- | --- | --- | --- | --- | --- |
| | | Mean | Median | Minimum | Maximum | |
| Aquifer Storage and Recovery | 19 | 12,583.90 | 11,356.24 | 624.59 | 53,942.12 | 239,094.18 |
| Brackish Groundwater | 119 | 14,980.88 | 11,356.24 | 37.85 | 113,562.35 | 1,782,724.54 |
| Groundwater Recharge | 20 | 13,979.53 | 10,712.72 | 832.79 | 37,854.12 | 279,590.51 |
| Other Non-Traditional Source and Projects [1] | 23 | 11,250.90 | 2839.06 | 113.56 | 57,159.72 | 258,770.75 |
| Reclaimed Water (for potable offset) | 377 | 8331.70 | 2536.23 | 18.93 | 124,161.51 | 3,141,049.52 |
| Stormwater | 23 | 16,598.21 | 9463.53 | 454.25 | 46,939.11 | 381,758.78 |
| Surface Water and Storage [2] | 64 | 35,264.78 | 18,927.06 | 7.57 | 463,334.40 | 2,256,945.78 |

[1] Combines "Other Non-Traditional Sources" and "Other Project Type" defined in [10]. [2] Combines "Surface Water" and "Surface Water Storage" defined in [10].

Reclaimed water (for potable offset), brackish groundwater, and surface water and storage were the most numerous project types, accounting for 86.8 percent of the projects in the database (Table 2). The narrative descriptions provided for the projects were summarized for illustrative purposes and displayed as WordClouds (see Figure 3). Reclaimed water (for potable offset) projects included a broad range of projects, primarily construction or expansion of transmission capacity to provide reclaimed water for urban irrigation. In fact, in 2015, out of 2.31 million m³ per day (or 609.0 mgd) of reclaimed water used to meet water demand in Florida (i.e., excluding reclaimed water for groundwater recharge), 68.9 percent was applied for residential irrigation, golf course irrigation, and other public access areas. Reclaimed water use for these three reuse activities had been growing steadily since 2000, while the reuse for agricultural irrigation had been declining (as summarized in [10] from the annual FDEP reuse reports [31]). Reclaimed water is expected to be a main source of alternative water supply for urban uses statewide.

Brackish groundwater projects were primarily described in the database as brackish groundwater development or well construction (Figure 3). Such projects were especially important for SFWMD, where this category accounts for the largest proportion of the projects, based on both the number and the total project capacity. Brackish groundwater was also important in the CFWI area; however, this source was less available in the other parts of the state.

Surface water and storage was the third most prominent project type in the dataset. Many surface water projects included recharge components (Figure 3), increasing aquifer recharge rates and benefiting both natural systems and groundwater users. Projects of this type were proposed or implemented in all WMDs, and their number and capacity was especially large in SWFWMD, where surface water and storage accounted for 44.7 percent of the capacity of all alternative water supply projects.

The other types included a relatively small number of projects, likely because of regional (as opposed to statewide) significance of the respective sources. In NFRWSP—the planning region shared by SRWMD and SJRWMD—groundwater recharge accounted for 65.48 percent of projects' total capacity. In turn, in NWFWMD, projects listed as "other non-traditional sources and projects" accounts for 47.04 percent of the entire capacity for all the listed projects. Almost all ASR projects were from SFWMD. Finally, stormwater projects were listed by all the WMDs (except NWFWMD); however, their total number and capacity was small.

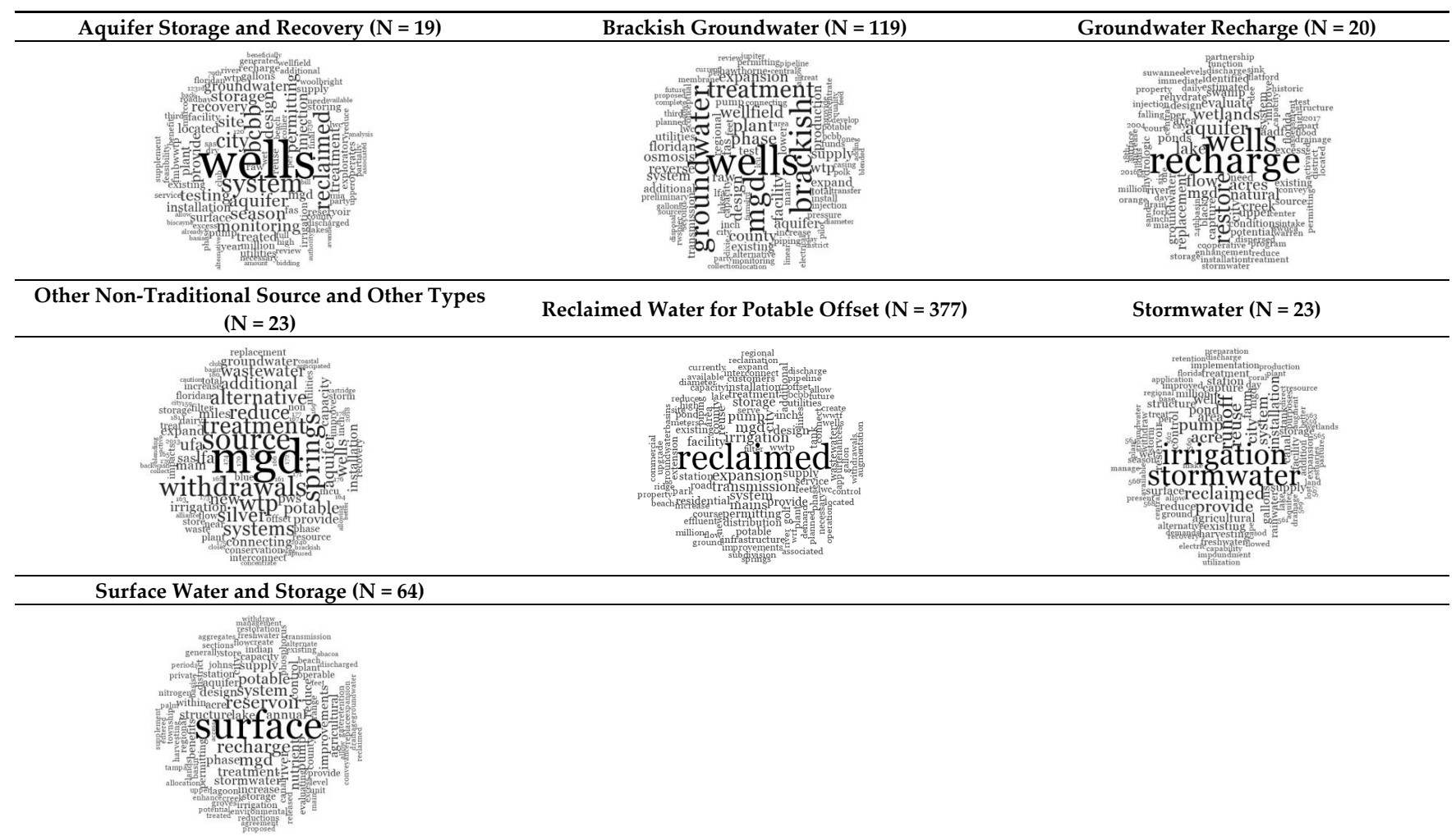

**Figure 3.** Project description: WordCloud summary, by project type (100 most common words; created in NVivo Software, version 12, using "with stemmed words" option).

The project number varied significantly by region. Given that existing supplies were projected to meet most of the demand increase in NWFWMD, the database included only 12 projects from the WMD (i.e., less than 2% of the database). We combined NWFWMD, SRWMD, and SJRWMD (outside CFWI) into the same region representing north Florida for the analysis below. In contrast, the fast-growing CFWI area accounted for approximately 20 percent of all projects in the database (Table 3). In terms of the project sizes, surface water and storage projects tended to be exceptionally large (Table 2). The projects of all types tended to be larger in SFWMD. Overall, regional variation in the project sizes and types was expected to impact the expenditures for satisfying the state's growing water demand.

**Table 3.** Project number, by geographical region.

| Geographical Regions | Number of Projects | Percent |
|---|---|---|
| NWFWMD, SRWMD, and SJRWMD [1] | 161 | 24.96 |
| CFWI | 132 | 20.47 |
| SWFWMD [1] | 182 | 28.22 |
| SFWMD [1] | 170 | 26.36 |
| Total | 645 | 100.00 |

[1] Excluding CFWI.

The majority of the projects (52.9%) had not yet been completed (Table 4), and therefore, they can provide a guide of potential future water supply sources in various regions. Based on the total project capacity for such projects, reclaimed water, brackish groundwater, and surface water and storage were expected to be keys to meeting the future water demand, with other project types also being prominent in specific regions (Table 5).

**Table 4.** Project number, size, and total capacity, by project types.

| Project Status | Number of Projects | Percent |
|---|---|---|
| Completed | 304 | 47.13 |
| Design, construction/underway, or on hold | 98 | 15.19 |
| RWSP/RPS Options Only | 243 | 37.67 |
| Total | 645 | 100.00 |

**Table 5.** Total capacity of the projects that had not yet been completed, by type and geographic region (million cubic meters per day).

| Project Type | Geographic Regions | | | | |
|---|---|---|---|---|---|
| | NWFWMD, SRWMD, and SJRWMD [1] | SWFWMD [1] | SFWMD [1] | CFWI | Total |
| Aquifer Storage and Recovery | 0.00 | 0.01 | 0.06 | 0.00 | 0.07 |
| Brackish Groundwater | 0.00 | 0.10 | 0.42 | 0.37 | 0.90 |
| Groundwater Recharge | 0.20 | 0.03 | 0.00 | 0.00 | 0.23 |
| Other Non-Traditional Source and Projects [2] | 0.11 | 0.00 | 0.04 | 0.00 | 0.15 |
| Reclaimed Water (for potable offset) | 0.08 | 0.80 | 0.39 | 0.14 | 1.41 |
| Stormwater | 0.04 | 0.19 | 0.04 | 0.00 | 0.28 |
| Surface Water and Storage [3] | 0.26 | 0.85 | 0.58 | 0.14 | 1.83 |
| Total | 0.69 | 1.98 | 1.53 | 0.66 | 4.86 |

[1] Excluding CFWI. [2] Combines "Other Non-Traditional Sources" and "Other Project Type" defined in [10]. [3] Combines "Surface Water" and "Surface Water Storage" defined in [10].

### 3.2. Regression Analysis of the Project Costs

The median project implementation costs were $3.54 million (with the mean of $24.01 million). The project cost increased with the project size; the natural logarithm of the implementation costs and the natural logarithm of the project capacity were highly correlated (Figure 4). The regression analysis showed that, as expected, the project implementation costs increased with the project sizes (Table 6). Further, the projects identified as options to meet the future increase in water demand (i.e., RWSP and RPS options) tended to be more expensive than the projects of the other statuses. This result could be due to the planning-level cost estimates provided for the future projects, or because the opportunities to implement the low-cost projects had been exhausted. Among water sources, stormwater projects were significantly less expensive as compared with surface water and storage projects (i.e., the benchmark). Finally, projects from north and central Florida (i.e., NWFWMD, SRWMD, and SJRWMD and CFWI) tended to be less expensive than the same projects implemented in SWFWMD (i.e., the benchmark region). Overall, the model explained approximately 54 percent of the variability of the independent variable.

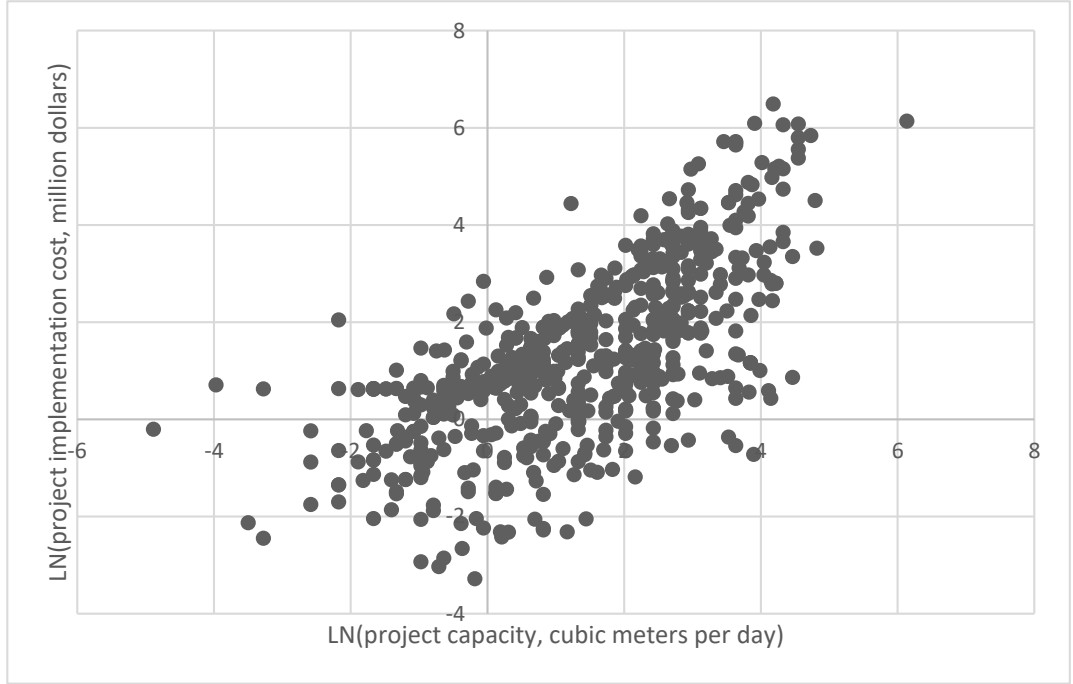

**Figure 4.** Scatter Plot of the Natural Logarithms of the Project Implementation Costs and Capacities.

**Table 6.** Estimated coefficients for the variables used in the regression model (with the coefficients that are statistically different from zero highlighted in bold font).

| Parameter | | Estimate | Standard Error | 95% Confidence Limits | | Chi-Square | Pr > $\chi^2$ |
|---|---|---|---|---|---|---|---|
| Intercept | | **−8.560** | 0.797 | −10.121 | −6.998 | 115.470 | <0.0001 |
| Natural logarithm of project capacity | | **1.256** | 0.083 | 1.093 | 1.419 | 227.700 | <0.0001 |
| Project Type | Aquifer Storage and Recovery | **6.051** | 2.036 | 2.060 | 10.042 | 8.830 | 0.003 |
| | Brackish Groundwater | **3.263** | 0.972 | 1.358 | 5.168 | 11.260 | 0.001 |
| | Groundwater Recharge | **−4.766** | 2.044 | −8.772 | −0.760 | 5.440 | 0.020 |
| | Other Nontraditional Sources and Project Types | **3.293** | 1.317 | 0.712 | 5.875 | 6.250 | 0.012 |
| | Reclaimed Water (for potable offset) | **3.053** | 0.788 | 1.508 | 4.597 | 15.010 | 0.000 |
| | Stormwater | **5.242** | 1.528 | 2.246 | 8.237 | 11.760 | 0.001 |
| | Surface Water and Storage (benchmark) | 0.00 | | | | | |
| Region | NWFWMD, SRWMD, & SJRWMD | **2.387** | 0.600 | 1.212 | 3.562 | 15.850 | <0.0001 |
| | CFWI | **2.246** | 0.605 | 1.060 | 3.431 | 13.780 | 0.000 |
| | SFWMD | −0.328 | 0.709 | −1.717 | 1.061 | 0.210 | 0.644 |
| | SWFWMD (benchmark) | 0.00 | | | | | |
| Interaction between project status and natural logarithm of project capacity | Completed | **−0.070** | 0.012 | −0.093 | −0.047 | 35.790 | <0.0001 |
| | Design, construction/underway, or on hold | **−0.074** | 0.016 | −0.106 | −0.043 | 20.960 | <0.0001 |
| | RWSP/RPS Options (benchmark) | 0.00 | | | | | |
| Interaction between project region and natural logarithm of project capacity | NWFWMD, SRWMD, & SJRWMD | **−0.368** | 0.074 | −0.513 | −0.222 | 24.420 | <0.0001 |
| | CFWI | **−0.334** | 0.073 | −0.477 | −0.190 | 20.770 | <0.0001 |
| | SFWMD | −0.057 | 0.079 | −0.213 | 0.099 | 0.510 | 0.473 |
| | SWFWMD (benchmark) | 0.00 | | | | | |
| Interaction between project type and natural logarithm of project capacity | Aquifer Storage and Recovery | **−0.674** | 0.226 | −1.116 | −0.231 | 8.910 | 0.003 |
| | Brackish Groundwater | **−0.313** | 0.104 | −0.516 | −0.109 | 9.080 | 0.003 |
| | Groundwater Recharge | **0.446** | 0.224 | 0.007 | 0.884 | 3.960 | 0.047 |
| | Other Nontraditional Sources and Project Types | **−0.341** | 0.152 | −0.639 | −0.043 | 5.020 | 0.025 |

**Table 6.** *Cont.*

| Parameter | Estimate | Standard Error | 95% Confidence Limits | | Chi-Square | Pr > $\chi^2$ |
|---|---|---|---|---|---|---|
| Reclaimed Water (for potable offset) | **−0.329** | 0.084 | −0.492 | −0.165 | 15.500 | <0.0001 |
| Stormwater | **−0.748** | 0.166 | −1.073 | −0.424 | 20.410 | <0.0001 |
| Surface Water and Storage (benchmark) | 0.00 | | | | | |

| | |
|---|---|
| R-Square | 0.539 |
| AICR | 830.192 |
| BICR | 930.593 |
| Deviance | 640.206 |

### 3.3. Regional Project Implementation Scenarios and Statewide Expenditure Projections

The relation between the project implementation costs and project capacity, type, and location was used to estimate the costs to implement the projects in various regions. In this analysis, we used the average capacity for each project type, with the assumption that the projects design or implementation had not yet started (i.e., project status is "RWSP/RPS Options"). The estimated project costs were then divided by the mean project capacity to assess the implementation cost per unit of the project capacity (Table 7). As discussed above, stormwater projects tended to be the least expensive option, while brackish groundwater was relatively more expensive. Also, projects implemented in SWFWMD tended to be more expensive, especially when compared with north Florida (i.e., NWFWMD, SRWMD, and SJRWMD).

**Table 7.** Estimated implementation costs, assuming mean projects capacity. [1]

| Project Types | Mean Project Capacity (Cubic Meters Per Day) | Implementation Costs (dollars Per Cubic Meter Per Day of Capacity) | | | |
|---|---|---|---|---|---|
| | | NWFWMD, SRWMD, and SJRWMD [2] | SWFWMD [2] | SFWMD [2] | CFWI |
| Aquifer Storage and Recovery | 12,583.90 | 534.71 | 1578.60 | 664.77 | 640.02 |
| Brackish Groundwater | 14,980.88 | 919.88 | 2895.41 | 1207.27 | 1107.58 |
| Groundwater Recharge | 13,979.53 | 429.96 | 1319.38 | 552.30 | 516.48 |
| Other NTS&PT | 11,250.90 | 826.83 | 2342.59 | 992.81 | 985.90 |
| Reclaimed Water (for potable offset) | 8331.70 | 829.20 | 2103.77 | 906.96 | 978.68 |
| Stormwater | 16,598.21 | 92.66 | 302.85 | 125.54 | 111.96 |
| Surface Water and Storage | 35,264.78 | 648.19 | 2794.61 | 1109.84 | 803.51 |

[1] Note that these costs include capital investment only, and do not account for operation and maintenance costs. While all these project types are expected to provide water for growing water demand, the water quality provided by the projects can differ; and therefore, the project capacity reported in this table should not be the only metrics to consider. [2] Excluding CFWI.

The statewide expenditures to meet the water demand increase in the next 20 years is presented in Table 8. The estimate utilizes the implementation costs reported in Table 7; we also assume that the mix of water supply sources used to meet the projected water demand should mimic the mix of various sources reported in Table 5. In other words, the share of various project types observed in the database (excluding the completed projects) is assumed to correspond to the share of the alternative sources in the future water supply mix in each region. The total statewide expenditures are projected to approach $2 billion over the next 20-year planning horizon.

**Table 8.** Estimated implementation costs to meet the increase in water demand in the next 20 years, million dollars [1].

| | NWFWMD, SRWMD, and SJRWMD [2] | SWFWMD [2] | SFWMD [2] | CFWI | Total Statewide |
|---|---|---|---|---|---|
| Total project implementation costs | 322.50 | 287.10 | 246.11 | 897.14 | 1752.85 |

[1] Note that these costs include capital investment only, and does not account for operation and maintenance costs. [2] Excluding CFWI.

## 4. Discussion

The analysis shows that over the next 20 years, Florida must invest almost $2 billion into the development of alternative water supplies to meet the growing water demand. On average, approximately 65 percent of expenditures for such water supply projects are funded by local partners. In comparison, regional and state funding shares account for about 35 and 5 percent of the total, respectively [17]. Therefore, the local entities must plan for these expenditures accordingly. These investments can also be particularly high for the urbanizing regions with significant population growth, requiring such communities to explore appropriate funding mechanisms.

The actual investments to meet the water demand increase can be even higher than reported in Table 8. The average beneficial offset provided by the reclaimed water projects is determined at 0.55 [17,32]. Such a beneficial offset ratio implies that approximately two cubic meters per day of water should be created by reclaimed water projects to meet the rise in the future water demand of one cubic meter per day. The offset values vary among the types of reuse activities. For example, the offset provided by reclaimed water for fire protection is estimated at one [32]. For various urban irrigation activities, the offset ratio is between 0.25 and 0.75 [32]. This low ratio reflects the poor irrigation efficiency pertinent specifically to reclaimed water use. This low ratio also captures an expansion of irrigated areas due to reclaimed water availability (e.g., irrigation of road medians if reclaimed water is available, but no irrigation otherwise) [32]. If the offset ratio is significantly lower than 1, the costs of reclaimed water projects needed to meet the future demand will dramatically increase.

Further, the expenditures reported in Table 8 include capital investments only and do not account for the operation and maintenance (O&M) costs. O&M expenses will further increase the total funding required to satisfy the water demand increase. This study also does not consider the expenditures for data collection and evaluation (such as water quality monitoring, or hydrogeologic modeling), which can also be expensive, and which are critical for selecting water resource use and management strategies [33].

This study relies on the demand projections that do not explicitly account for the potential demand increase during droughts. For the last year of their planning horizons, Florida's WMDs forecast the water demand for both normal and drought rainfalls. The drought event is defined as "a year in which below normal rainfall occurs with a 10 percent probability of occurring in any given year" [19] (p. 4). Drought demand coefficients, historical water use analysis, and crop irrigation requirement models are used to develop the drought demand forecast [34]. The compilation of WMDs' most recent drought demand projections is not available; however, a 2018 compilation shows that the statewide drought demand can exceed the demand given normal rainfall by approximately 20 percent [34]. Such a significant increase in water demand during droughts and potential impacts of the drought on water supply can require additional investments in the alternative water supplies.

At the same time, considerable opportunities exist in the state to reduce funding needs for alternative water supplies. First, implementation costs differ significantly among the water supply sources, and more substantial utilization of inexpensive sources, such as stormwater, can reduce the regional and statewide cost. For example, stormwater is expected to provide approximately 3 percent of the water to meet future demand (based on the project database utilized in this study). If, in every region, stormwater projects are used to satisfy 10 percent of the future water demand (with a corresponding reduction of surface water and storage), the total statewide implementation costs become $1662.11 million over 20-year period, or 5.18 percent lower than reported in Table 8.

Second, the statewide expenditure can be reduced by varying the sizes of the projects implemented to meet the future water demand, if feasible. The regression model shows a statistically significant effect of the project size on the project implementation costs, even though a massive increase in the project sizes is needed to noticeably alter the statewide expenditure estimate. For example, for a 10 percent increase in the project size assumed for each project type, the estimated total statewide cost reduces by only 1.73 percent.

Third and most importantly, water conservation can provide a cost-effective alternative to water supply development. This study utilizes the water demand projections that do not account for continuous water use efficiency improvement. Meanwhile, in the public supply sector, the statewide average per capita use declined sharply in 2000–2010 and then remained unchanged in 2010–2015 [13]. While no statewide data are available for the 2015–2020 period, potential reductions in the per capita use are discussed in all RWSPs. To assess the offset for the alternative water supply costs that the water conservation can provide, we compared the water conservation potential, demand increase, and existing supplies for each of the water supply planning regions identified in Figure 1. Conservation potential can offset the requirement to develop water supplies for all planning regions except the Northern region in SWFWMD, as well as CFWI and NFRWSP. The remaining additional water supply needs for these three planning regions is 987,235.39 cubic meters per day (or 260.80 mgd). In other words, water conservation can reduce the necessity of developing alternative water supplies by 44.68 percent. The statewide water supply development investments for this scenario become $931.53 million over 20 years, showing more than 45 percent reduction in the water supply development costs as compared with Table 8.

Water conservation can be a more cost-efficient strategy for addressing the projected gap between water demand and existing supplies than developing alternative water supplies. For example, in central Florida, 11 active water conservation programs for the public supply category are considered, including such strategies as high-efficiency toilet installations, residential low-flow showerheads, irrigation and landscape evaluations, residential irrigation controllers, and irrigation enforcements. These 11 programs are estimated to result in 17,185.77 cubic meters per day (or 4.54 mgd) of water use reduction, costing $8.1 million over the 20-year planning horizon [21], or approximately $475 per m$^3$ per day (or $1.8 million per mgd). These average costs are significantly lower than that for widely used brackish groundwater or reclaimed water alternative water supply sources (see Table 8). Note that as mentioned above, the project database used in this study initially included 358 projects classified as "public supply and commercial-industrial-institutional conservation." For these projects, the median costs were $1207.27 per cubic meter per day of water conserved (or $4.57 million per mgd), which is relatively high and comparable to the costs of alternative water supply projects. These high costs are likely because the low-cost conservation initiatives, such as conservation water pricing or outreach strategies, are not eligible for WMDs' cost-share funding and are not included in the project database. Ultimately, both "rethinking water supply" and "rethinking water demand" [12] (p. 21302) should be considered when addressing water scarcity challenges.

## 5. Conclusions

In 2010, Dr. Peter H. Gleick, co-founder and director of the nonprofit research think-tank Pacific Institute, identified rethinking water supply and demand, reforming water management institutions, and analyzing climate change impacts as the solutions to global water crisis [12]. These suggestions reaffirm and extend the proposals expressed in other manuscripts (e.g., [35–37]), and they remain relevant in 2020. Moreover, the recommendations become applicable to even the historically water-rich regions like the eastern United States.

In this paper, we estimate the capital investments for developing new water supply sources alternative to the traditionally used groundwater in one of the largest and fastest-growing states in the United States – Florida. In line with Gleick's recommendations, Florida has been rethinking water supply and developing sources that have been ignored in the past, such as reclaimed water, brackish groundwater, surface water storage, and stormwater. Based on a database of projects assembled by FDEP, this study shows that over the next 20 years, Florida must invest almost 2 billion dollars into the development of alternative water supplies to meet the growing water demand. The expenditures projections depend on the mix of projects considered, with scenarios relying on such commonly used water sources as reclaimed water and brackish groundwater being relatively expensive. Less expensive project options include groundwater recharge and stormwater projects.

The scarcity of traditional water supply and the high costs of alternative water supply projects call for combining "rethinking water supply" with "rethinking water demand" [12] (p. 21302). In Florida, water conservation is defined as reduction in wasteful and inefficient water uses [19]. Water conservation can reduce the need for alternative water supplies by more than 40 percent, based on WMDs' estimates. This drop can translate into a reduction of more than 45 percent in the capital investments for alternative water supplies. While water conservation programs considered by the WMDs also require investments, many of them are less expensive (per m$^3$ or mgd of water) than the alternative water supply projects.

The solutions of "rethinking water supply" and "rethinking water demand" should be integrated and coordinated (e.g., [37]). Gleick suggests that "a key to improving efficiency is understanding where, when, and why we use water" [12] (p. 21302). A better understanding of our water use may allow tailoring the water supply treatment levels to the water use requirements. This adaptation of supply sources has been partially accomplished by separating the water applied for irrigation and providing reclaimed water specifically for this use. More specific water quality requirements for different use types may be developed in the future, with various supply mixes appropriate for each application type. In this case, the statewide total water demand projections may become irrelevant, as the demand, supply, and expenditure projections will need to be developed for each type of use. Reduction in the expenditures could be possible if treatment needs vary significantly among the use types, with many uses requiring reduced treatment levels.

Gleick [12] also emphasizes climate change implications for water resource management decisions. Potential impacts of climate change on both water demand and supply sources are discussed in Florida's RWSPs; however, no quantitative assessment is available. Similarly, this study includes no analysis of establishing better institutions to address potential groundwater quality or the implications of storage, recovery, and recharge activities [38].

Overall, climate change impacts, natural systems restoration needs, and maintenance and replacement requirements for the aging infrastructure will make the total expenditures far exceed the estimates reported in this study [17]. For example, the United States Environmental Protection Agency's (US EPA's) Drinking Water Infrastructure Needs Survey provides the most comprehensive infrastructure cost estimate in the United States [39]. The US EPA survey focuses on existing infrastructure maintenance, and it largely excludes new infrastructure necessary to meet the increasing demand due to future growth. The study estimates that $21.89 billion is needed for Florida's existing infrastructure in the next 20 years [39]. The expenditures to continue providing water to communities and to support future growth are real needs for many U.S. states and water providers [40]. Overall, expenditure analysis should continue to be an essential part of the long-term water supply planning and water resource management.

**Author Contributions:** Conceptualization, T.B., M.C., K.B., and K.H.; methodology, T.B.; formal analysis, T.B.; Writing—Original draft preparation, T.B.; writing—review and editing—original draft, M.C., K.B., and K.H.; Writing—Review and Editing—Revised versions of the manuscript, M.C., and K.B.; supervision, T.B. and M.C.; project administration, M.C. All authors have read and agreed to the published version of the manuscript.

**Funding:** This study is partially funded through the cooperative agreement between the Office of Economic and Demographic Research, Florida Legislature, and the University of Florida. Any opinions, findings, conclusions, or recommendations expressed in this publication are those of the author(s). They do not necessarily reflect the view of Office of Economic and Demographic Research, Florida Legislature. This paper is also partially supported by the U.S. Department of Agriculture Project FLA-FRE−005565 (PI: Tatiana Borisova).

**Acknowledgments:** We appreciate insightful feedback provided by two anonymous reviewers that allowed us to improve the manuscript significantly. We also valuable acknowledge assistance from the professional editor, Ms. Carol Fountain (Food and Resource Economics Department, University of Florida). Finally, we wanted to thank Dr. James Colee, Consultant, Statistical Consulting Unit, Institute of Food and Agricultural Sciences, University of Florida, for constructive discussions about the methods used in this study.

**Conflicts of Interest:** The authors declare no conflict of interest.

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
