# Peer review of "Addressing the Scarcity of Traditional Water Sources through Investments in Alternative Water Supplies: Case Study from Florida"

_water, doi:10.3390/w12082089_

Round 1
Reviewer 1 Report
Attached MS Word document

Author Response
Dear Reviewer,
Thank you for taking your time and providing helpful recommendations for our manuscript! We revised the document according to your suggestions.
The summary of the revisions is included in the attached file.
Best regards,
Authors

Reviewer 2 Report
row 32. Demand management is poorly described, as you point out on r398. Do the project reports you cite contain any such info? Do spome of them make the distinction between technical devices to reduce consumption levels and management issues such as progressive tariffs and short shower baths?
Are some demand mgnt measures already included in your "conservation"? This is not clear in your statements on r151. It becomes important to know as a basis for your important statement r256 that such measures are relatively expensive.
Reclaimed water, as defined in r133-135, may include any used water. The cost for such project must differ hugely depending of the resulting water being potable or non-potable.
r112. The reader would benefit from data on anticipated population increases over the period already in the introduction. Also, to have data on what proportion of supplied water today goes to households, CII, agriculture, nature, restauration etc. Such info would help to understand why (if this is true) that low-flush toilets have little impact on demand. My understanding is that it may save 50-60 lt pcd.
When costs are discussed (Table 4) they relate to capital investment r256. There is no O&M cost comparison. This makes cost conclusions vague. Also, the quality of the incoming water, and more so the required quality of supplied potable water, affects the cost of projects. Thus, the data in Table 4 should be qualified more from these aspects.
These points on cost reflects on Table 6 and 7. Experimental projects are considerably cheaper than the conventional ones. But the conventional projects comprise potable water offset. Do we compare apples and pears or have I misunderstood the text?
Author Response
Dear Reviewer,
We appreciate the time and effort that you dedicated to providing feedback on our manuscript, and we are grateful for the insightful comments on our paper.
We have incorporated most of the suggestions. Please see the detailed responses to your recommendations in the attached file.
Sincerely,
Authors

Round 2
Reviewer 1 Report
Thanks for responding to my suggestions. This version is much improved.
Author Response
Thank you! Your insightful comments (in the previous review round) helped us improve our paper.
Reviewer 2 Report
The revisions made are commendable. Figure 2 is excellent in this paper.
Now, it is possible to give some more suggestions how to improve the text further.
a) The title mentions about ´investments in Alternative water supply´, which is fine. But the alternatives are missing in the Abstract itself.
b) The introduction mentions that per capita use has shrinked and caused financial problems for the utilities. The population is expected to increase 20% in the 20 year period. The reader is eager to know whether the demand-supply gap could be filled with improved efficiency in the existing supply. The answer on row 99 (R99) is that the demand management is not included in the projections, and per capita use remains unchanged. This statement should be addressed in the discussion or conclusion.
R122-124 gives the reason that estimates of efficiency improvements has a high degree of uncertainty. That is true but not much more uncertain than all the other parameters. Since WMD are likely to prefer to build new infrastructure, researchers must be more impartial and discuss demand management issues and reuse seriously. We owe that to the public. R256-260 tells that you excluded ”data collection and evaluation” from the project list because these did not directly contribute to the development of new water supplies. Do you not run the risk of excluding demand management measures since you do not even evaluate the total situation?
c) In order to facilitate the reading, I suggest the following shifts:
Figure 2 to be presented before Table 1.
R83-88 are moved to R95 after … (see Figure 2). As a new paragraph.
R89 Figure 2 shows that the most substantial increase in demand (+651 mgd) is … and the mgd are also mentioned for the other uses.
Before Table 1 we need a definition of Conservation (R117-124). Then your definition of Alternative water supply (R145-149 ) should be moved here too.
Table 1 has some problems with numbers. 1.4 should read 20.4; 140.2 should read 101.7; 62.6 should read 323.2; and 471.5 should read 718.5 (if the other figures are correct).
Directly after Table 1, you should include some comments, like SFWMD has the biggest gap, and only 1/3 can be covered by conservation. In CFWI, only little can be remedied through conservation and Alternative supplies are called for. Etc.
R143 62.7 mgd should be 323.2
R147 there is no definition of reclaimed water here. Important to tell whether this is ”recycled water”, ”new water” or other often used definition.
R273 Reclaimed water is defined as urban irrigation, which is reasonable if used water is used for this purpose and therefore free potable water for urban usage. But, how can expansion of transmission capacity belong to this group? I am not aware that. Why should e.g. used urban water used to water parks require twice the volume compared to council water. Needs an explanation, since it has a huge implication on cost.
Table 4 heading ”Total sample” should read Capacity in mgd
Table 6. This table giving capacity of projects in the pipeline to cover the demand-supply gap is important. I suggest the data is discussed in relation to Table 1. For instance, CFWI has three kinds of conservation projects adding up to 173.16 mgd, while Table 1 tells that only 36.8 mgd is provided. Or have I misunderstood the meaning of column ”Alternative water supply or conservation”?
R342 Can medium and mean differ that much?
Table 8. Stormwater projects outdo all other measures, whereas the other project types are similar in cost (with few exceptions). One general question is that if the project types provide the same level of water quality? If not, are the treatment costs included? You write in other section that O&M cost are left out. If not included, the Table becomes misleading.
R415 471.5 should read 718.5 mgd if I have understood Table 1 correctly. The addition reaches 450 mgd.
Author Response
Please, find the response in the file attached.
